# Dysbiosis of Gut Microbiota from the Perspective of the Gut–Brain Axis: Role in the Provocation of Neurological Disorders

**DOI:** 10.3390/metabo12111064

**Published:** 2022-11-03

**Authors:** Meenakshi Kandpal, Omkar Indari, Budhadev Baral, Shweta Jakhmola, Deeksha Tiwari, Vasundhra Bhandari, Rajan Kumar Pandey, Kiran Bala, Avinash Sonawane, Hem Chandra Jha

**Affiliations:** 1Infection Bioengineering Group, Department of Biosciences and Biomedical Engineering, Indian Institute of Technology Indore, Simrol, Indore 453552, Madhya Pradesh, India; 2Department of Pharmacoinformatics, National Institute of Pharmaceutical Education and Research, Hyderabad 500037, Telengana, India; 3Department of Medical Biochemistry and Biophysics, Karolinska Institute, 17165 Stockholm, Sweden; 4Algal Ecotechnology & Sustainability Group, Department of Biosciences and Biomedical Engineering, Indian Institute of Technology Indore, Simrol, Indore 453552, Madhya Pradesh, India; 5Disease Biology & Cellular Immunology Lab, Department of Biosciences and Biomedical Engineering, Indian Institute of Technology Indore, Simrol, Indore 453552, Madhya Pradesh, India

**Keywords:** gut–brain axis, neurodegenerative disease, gut microbiota, gut dysbiosis, vagus nerve, neuroinflammation

## Abstract

The gut–brain axis is a bidirectional communication network connecting the gastrointestinal tract and central nervous system. The axis keeps track of gastrointestinal activities and integrates them to connect gut health to higher cognitive parts of the brain. Disruption in this connection may facilitate various neurological and gastrointestinal problems. Neurodegenerative diseases are characterized by the progressive dysfunction of specific populations of neurons, determining clinical presentation. Misfolded protein aggregates that cause cellular toxicity and that aid in the collapse of cellular proteostasis are a defining characteristic of neurodegenerative proteinopathies. These disorders are not only caused by changes in the neural compartment but also due to other factors of non-neural origin. Mounting data reveal that the majority of gastrointestinal (GI) physiologies and mechanics are governed by the central nervous system (CNS). Furthermore, the gut microbiota plays a critical role in the regulation and physiological function of the brain, although the mechanism involved has not yet been fully interpreted. One of the emerging explanations of the start and progression of many neurodegenerative illnesses is dysbiosis of the gut microbial makeup. The present understanding of the literature surrounding the relationship between intestinal dysbiosis and the emergence of certain neurological diseases, such as Alzheimer’s disease, Parkinson’s disease, Huntington’s disease, and multiple sclerosis, is the main emphasis of this review. The potential entry pathway of the pathogen-associated secretions and toxins into the CNS compartment has been explored in this article at the outset of neuropathology. We have also included the possible mechanism of undelaying the synergistic effect of infections, their metabolites, and other interactions based on the current understanding.

## 1. Introduction

Microbial diversity colonizes nearly every nook and corner of the human body accessible to outer surroundings, including the skin as well as the gastrointestinal, genitourinary, and respiratory tracts. The number of microbial entities in the human body is believed to be ~10^13^–10^14^, which corresponds to a 1:1 ratio of human and microbial cells [1]. This consortium of microbial communities resides in a specific biological niche, collectively known as the “microbiota” [2]. The human microbiota includes a variety of fungi, viruses, archaebacteria, protozoa, and, predominantly, bacteria [3]. Although some are toxic, most microbial diversity is innocuous or even helpful to humans. These microorganisms residing inside/upon the human body play a remarkable role in maintaining the homeostasis of human health [4]. The human microbiota contributes significantly to developing the immune system and to building a physical barrier against pathogenic organisms [5]. However, microbial homeostasis perturbations also correspond to various diseased states, for example, inflammatory bowel disease (IBD) [6], Crohn’s disease (CD) [7], Alzheimer’s disease (AD) [8] and Parkinson’s disease (PD) [9], non-alcoholic fatty liver disease [10], atopic eczema allergies [11], gastric cancer [12], etc. Therefore, it is of the utmost importance to investigate the function of the microbiota in normal physical and pathological conditions to aid the development of therapeutics against them.

Furthermore, the discovery of microbial diversity in different anatomical locations has led to the detailed study and characterization of the microorganisms inhabiting varied microenvironments in the human body [13]. The Human Microbiome Project (HMP) is one such initiative taken up by the Common Fund-NIH to explore the job of gut microbiota in human health and diseases [14]. The term “microbiome” refers to a collection of microbial populations that inhabit a specific environment, including their ecosystem, collective genome, and the surrounding environmental conditions [15]. The HMP has analyzed the microbial communities inhabiting various human body niches, such as the skin, oral cavity, gastrointestinal tract, and urogenital tract, and their role in healthy and diseased states [16].

The gut microbiota aids in a wide range of functions in human health; due to its prominent role in the neuroendocrine functions [17], metabolism [18], and immunity of an individual [19], the gastrointestinal microbiome has piqued the scientific community’s curiosity in recent years [20]. The gastrointestinal tract (GIT) is one of the most heavily colonized organs. The human gut has the largest surface area and is rich in nutrients that bacteria can use as a substrate, making it an ideal location for colonization [21]. Over 70% of all microorganisms in the human body are thought to reside in the colon.

The role of the gut microbiota in regulating various neuroendocrine functions has been widely studied [22]. However, there is increasing agreement among human and animal studies that the disruption of the gut microbiota impacts brain development, neurological outcomes, and disorders, resulting in long-term behavioral changes [23]. Hence, gut dysbiosis is reported to be associated with several neurological disorders, including Alzheimer’s [24], Parkinson’s [25], Huntington’s disease (HD) [26], and multiple sclerosis (MS) [27].

## 2. Composition of Gut Microbiota and Its Associated Multifarious Function in Host Heath

The human gut microbiota weighs approximately 2.1 kg and contains about 50–100 times more information than the human genome, making it a distinct organ of the human body [28]. These bacteria predominantly live on the human body as symbionts, assisting the host in maintaining homeostasis in steady-state conditions. The gut microbiota makes significant contributions to (i) gut permeability; (ii) preventing pathogen colonization and invasion; (iii) facilitating nutrient metabolism; (iv) assisting in the synthesis of vitamins such as vitamin K, vitamin B complexes, and folate; (v) expediting essential intestinal epithelial roles such as absorption and secretion [29]; and (vi) regulating innate and acquired immune responses that locally (GI mucosa level) and systemically influence distant organs [30].

*Firmicutes*, *Bacteroidetes*, *Proteobacteria*, *Actinobacteria*, *Verrucomicrobia*, and *Fusobacteria* comprise the predominant gut microbial diversity [31]. Ninety percent of the microbial population comprises *Firmicutes* and *Bacteroidetes*. *Bacteroidetes* constitute prominent genera such as *Bacteroides, Parabacteroides,* and *Prevotella* [32]. *Firmicutes*, the other dominant group of bacteria, includes several genera, such as *Clostridium*, *Lactobacillus*, *Streptococcus*, *Enterococcus*, *Eubacterium*, and *Ruminococcus*, among others [33]. *Bifidobacterium* and *Collinsela* are two significant genera of the phylum *Actinobacteria*, accounting for less than 10% of the total gut microbiota. *Helicobacter* and *Escherichia* are the dominant genera of *Proteobacteria* (less than 2%). *Fusobacterium* and *Akkermansia* dominate the genera of the phyla *Fusobacteria* and *Verrucomicrobia*, respectively, and constitute less than 3% of total gut microbial diversity [34].

### 2.1. Metabolic Function—Nutrient and Other Dietary Component Metabolism

The gut microbiota has been linked to various GIT activities, notably nutritional needs, energy storage, and diverse physiological pathways [35]. Dietary carbohydrates impart a significant portion of the microbiota’s nutrition. Short-chain fatty acids (SCFAs) are produced in the colon by bacteria such as *Bacteroides*, *Bifidobacterium*, *Roseburia*, *Faecalibacterium*, and *Enterobacteria* [36]. SCFAs such as butyrate, propionate, and acetate provide tremendous energy to hosts [37]. *Bacteroides* species are the most predominant organisms engaged in carbohydrate metabolism, as they encode for specific classes of enzymes, such as glycosyltransferases, glycoside hydrolases, and polysaccharide lyases, among others (Figure 1). For example, *Bacteroides thetaiotaomicron* has a genome that codes for over 260 hydrolases, significantly more than the human genome [38].

*Bacteroides* have been demonstrated to synthesize conjugated linoleic acid, a fatty acid possessing anti-diabetic, hypolipidemic, anti-obesogenic, immunomodulatory, and anti-atherogenic properties. A study published by the Velagapudi group showed that gut microbiota performed a crucial role in the regulation of lipogenesis when compared to conventionally raised mice (CONV-R) and germ-free mice (which lack a whole microbial community) [39]. The study reported higher levels of microbial-derived metabolites (citric acid, propionic acid, and fumaric acid) and lowered triglycerides and cholesterol levels in the serum metabolome of CONV-R mice compared to the metabolome of germ-free mice, thus demonstrating the effect of gut microbial diversity on host energy and lipid metabolism [40] (Figure 1). Furthermore, the gut microbiota has been associated with a reduced risk of renal oxalate stone formation. Because the production of oxalate in the intestine is a byproduct of bacterial metabolism and carbohydrate fermentation, it is quelled by organisms such as *Lactobacillus* species, *Bifidobacterium* species, and *Oxalobacter formigenes* [41].

The gut microbiota has been considered a cornerstone of maintaining the health status of its human host because it not only facilitates the harvesting of nutrients and energy from ingested food, but also produces numerous metabolites that can regulate host metabolism [42]. One such class of metabolites is the bile acids. Primary bile acids (BAs), such as cholic acid (CA) and chenodeoxycholic acid (CDCA), which are by-products of cholesterol metabolism and clearance, are synthesized by the liver [43] and are subsequently modified by the gut microbiota. The gut microbiota transform these compounds into an assortment of forms that significantly expand their biological value and diversity [44]. Primary bile acids have been reported to be modified by the gut microbiota in four distinct ways, including via the deconjugation of taurine or glycine amino acids as well as via the dehydrogenation, dihydroxylation, and epimerization of the cholesterol core [45]. A novel class of “microbially conjugated bile acids”, which have the ability to conjugate amino acids to bile acid, is synthesized by our gut bacteria in addition to the previously reported alterations [46].

### 2.2. Vitamins Supplier—Bacteria as a Source of Vitamins for Their Hosts

Among numerous associated functions, one of the beneficial activities of microorganisms is the production of multiple vitamins [47]. Another critical metabolic activity of the gut microbiota is the biosynthesis of vitamin K and various components of vitamin B (Figure 1). *Lactobacilli* appear to be unable to synthesize folate de novo, but some *Bifidobacterial* species belonging to the phylum *Actinobacteria* can synthesize folate [48]. *Firmicutes* and *Proteobacteria* species, *Bacillus subtilis*, and *Escherichia coli* can produce riboflavin [49]. Vitamin B12 is commercially produced utilizing *Propionibacterium freudenreichii*, a member of the phylum *Actinobacteria* [50]. Another *Firmicutes* phylum member, *Lactobacillus reuteri*, is thought to possess the potential for vitamin B12 production [51].

### 2.3. Immunomodulatory Role of Gut Microbiota

The gut microbiota helps to influence both the innate and adaptive immune systems in the gut [52]. The effector and regulatory T cells (Tregs), gut-associated lymphoid tissues (GALT), group 3 innate lymphoid cells, IgA-producing B (plasma) cells, dendritic cells in the lamina propria, and resident macrophages are immune response constituents and cell types that actively engage in the immunoregulatory process [53]. The gut microbiota is also necessary for Foxp3^+^ Treg cell development, differentiation, and function. However, the mechanism by which this task is facilitated is yet to be explored. For example, in specific *Clostridium* clusters, Treg induction may be independent of pattern recognition receptors (PRRs) or based on MyD88-dependent mechanisms [54], while in the case of *Bacillus fragilis*, the process appears to be mediated by Toll-like receptor-2 (TLR-2) signaling by polysaccharide A [55]. Furthermore, SCFAs derived from microbial metabolism, particularly butyrate, have indeed been coupled with Treg development and function [56]. SCFAs have been visualized to stimulate G-protein-coupled receptors by IECs and have been shown to govern Tregs via epigenetic regulation (enhanced acetylation) of the Foxp3 locus [57] (Figure 1).

## 3. Host Factors Controlling Gut Microbiota

The host selects its gut microbiota by generating various molecular signals and effector molecules that regulate the framework of microbiota-colonized surfaces and thus influence their composition [58]. Numerous associated host factors regulate the constitution of gut microbiota, and some of these factors are mentioned below.

### 3.1. Delivery Pattern

The mode of delivery is regarded as an essential influential factor in developing the gut microbiota [59]. The composition and architecture of the gut microbial community vary significantly among infants born via cesarean section and those born vaginally [60]. The maternal/vaginal microbiota is denied to Caesarean-born babies, and the first exposure is marked by the absence of strict anaerobes and facultative anaerobes such as *Clostridium species* [61]. After birth, the delivery method influences gut microbial development in the early stages of life. *Prevotella* and *Lactobacillus* from the mother’s vaginal microbial community predominate the foremost gut microbiota of newborns delivered vaginally [62]. On the other hand, those born via cesarean section receive their gut microbiota from their skin, which tends to result in the dominance of the following microbial populations: *Streptococcus, Corynebacterium*, and *Propionibacterium.* [63].

### 3.2. Infant Feeding

Infant feeding is another crucial factor that regulates gut microbial assembly, as the mother’s milk is rich in different prebiotic microorganisms that shape the infant’s gut microbiota [64]. More than 700 species of bacteria are present in a planktonic state in milk that maintains a balance with the immune cells present in milk [65]. Though the microbial diversity varies with lactation time [66], *Streptococcus*, *Staphylococcus*, *Pseudomonas*, and *Acinetobacter* are the predominant species found in breast milk. The average bacterial load is ~10^6^ microorganisms/mL, which increases by about 100 times as the infant receives 8 × 10^8^ bacterial cells with each 800 mL of breast milk consumed [67]. Colostrum samples taken at 1 and 6 months show a different diversity of bacteria, including *Prevotella*, *Leptotrichia*, and *Veillonella,* along with other microorganisms [68]. During breastfeeding, the skin interface allows some bacteria to become mixed with healthy milk microbiota [69]. The milk microbiome plays a vital role in infant growth and development. Obese mothers have lower diversity and a distinct microbiota composition in their breast milk compared to normal-weight mothers [70]. Immune cells that are interchanged through feeding, such as IgA, are essential for the inert immunity of infants [71].

### 3.3. Medication: Antibiotics

Antibiotics are a two-edged sword, as their inappropriate utilization kills both morbid and beneficial microbial flora indiscriminately, permitting dysbacteriosis and the growth of undesired microbes [72]. Antibiotic therapy diminishes the overall diversity of the intestinal flora, including the deprivation of some essential taxa, resulting in metabolic shifts, increasing the susceptibility to colonization, and the evolution of antibiotic-resistant bacteria [73]. In adults, a combination of gentamicin, meropenem, and vancomycin increased the popularity of *Enterobacteriaceae* and certain other pathobionts while decreasing the prevalence of *Bifidobacterium* and butyrate-producing organisms [74]. The inappropriate intake of antibiotics disrupts the crucial mechanism of the intestinal flora’s “competitive exclusion strategy,” [75] which exerts nutrient competition, niche exclusion, and competitive metabolic interaction to hamper the growth of pathogenic microbial diversity [76].

### 3.4. Genetics: mi-RNAs

Small non-coding RNA and mi-RNAs are another essential host factor that might be used to appraise the intestinal microbiota’s composition, shape, and structure [77]. Micro RNAs are single-stranded, short-length (approximately 18–23 nucleotides) endogenous RNA molecules that are synthesized within the nucleus and that are carried out to the cytoplasm afterward to regulate gene expression (gene silencing) [78]. As specific mi-RNA has a high degree of sequence complementarity with the target mRNA, mi-RNA hybridizes with the 3′ untranslated region of the target gene, facilitating mRNA degradation or impeding translation [79].

Liu et al. demonstrated that inter-species gene regulation via fecal microRNAs (miRNA) facilitates host control of the gut microbiota [80]. Fecal miRNA is predominantly synthesized by Hopx-positive cells and intestinal epithelial cells, as these two cells are found to secrete mi-RNA-containing exosomes. A direct correlation between mi-RNA-deficient IEC and associated gut dysbacteriosis has been studied in detail, and how fecal transplantation can rehabilitate intestinal flora has also been addressed as well, suggesting the potential role of mi-RNA in the regulation of the gut microbiota. Mi-RNAs can enter gut microbial cells; target bacterial genes precisely; and govern the growth, microbial gene transcripts, and prevalence of a particular gut microbiota [81]. Evidence that miRNAs may play a role in gut bacterial development comes from the improved in vitro growth of the bacteria *Fusobacterium nucleatum* and *E. coli* via hsa-miRNA-515-5P and hsa-miRNA-1226-5p, respectively. This was accomplished by cultivating the strains using synthetic miRNAs [82]. The administration of fecal miRNAs has effects that shape the gut microbiota [83].

## 4. Microbiota–Gut Brain Interconnection

The gastrointestinal tract and brain are interconnected via the paracrine effect of the signaling molecules secreted in the gut and vice versa. This reciprocal transmission of messages between the gut and brain, thereby performing functional regulation, is termed the “Gut–brain axis” (GBA) [84]. The GBA allows information to be exchanged back and forth between the digestive system and the central nervous system [85]. Interestingly, the gut microbiota and its secretions may modulate brain physiology by governing numerous processes such as development, aging, maturation, homeostasis, and various brain functions [86]. Brain behavior is directly impacted by microbial secretions [87]. According to the research on germ-free (GF) mice, the gut microbiota synthesizes many metabolites in the bloodstream; however, most of them are then modified by the host. These substances have a significant impact on mammalian behavior and neuroendocrine responses [88]. The microbes can influence neural activity by modulating various cerebral biochemical signaling pathways, potentially triggering cognitive impairments [89], thus suggesting their significant impact on brain functioning and physiology. Recent studies have focused on the intellectual effect of gut-on-brain and vice versa. The gut microbial community is vital for proper brain growth and mature cognitive function. The microbiota has been known to regulate critical neuroplasticity-related mechanisms in the adult brain, such as neurogenesis [90] and microglial activation [91]. The brain, on either hand, can alter the composition and different activities of the gut microbiota, such as its role in food absorption and metabolism [92]. Through the autonomic nervous system, the brain appears to significantly influence the structure, composition, and activity of the microbial community in the intestine. It can potentially influence regional gut motility, microbial gene expression via luminal hormone secretion, intestinal transit and secretion, and gut permeability [93]. Previous studies from our research group have demonstrated the pivotal role of pathogenic co-infection in making a conducive environment in the succession of multifactorial diseases such as cancer and parasitic diseases such as malaria [94]. Our investigations have corroborated previous reports that exposure to a bacterium (*H. pylori)* is associated with EBV reactivation, ultimately contributing to the aggravation of cancer pathology. When co-cultured with EBV, the I10 strain of *H. pylori* resulted in the elevated expression of various bacterium-associated pathogenic genes such as cagA and babA [95]. At the same time, EBV-associated lytic (gp350 and bzlf1) and latent (ebna1, ebna3c, lmp1, lmp2a, and lmp2b) genes were also found to be elevated in co-infected cells compared to single pathogen infection. This study provides evidence that co-infection creates a favorable microenvironment that promotes the growth of both pathogens, synergistically contributing to exacerbating disease severity [96]. Similarly, one pathogen may promote the production of another pathogenic microbe’s morbific gene, resulting in disease pathology aggression. Therefore, we took up the task of summarizing the consequences of gut microbial diversity on the brain and vice versa, focusing on pathogens such as *Helicobacter pylori* and Epstein–Barr virus. The microbiota–gut–brain axis is an arising and evolving concept for studying the influence of two distantly separated biological systems on one another [97]. This new discipline focuses on the induction, etiology, and progression of various metabolic and mental dysfunctions. Hence, understanding the microbial diversity of the gut–brain axis could also be utilized to develop novel therapeutic strategies to overcome neurological disorders. Therefore, in the current article, we mainly focused on reviewing the underexplored synergetic effects of pathogens and their interactions on the onset and progression of numerous pathologies, including cancers and neurological disorders. Additionally, the present article has considered the outcome of various gut microbial secretions in the advancement and modulation of AD, MS, PD, and HD.

## 5. Routeways for the Bidirectional Communication between the Gut and Brain

The GBA allows multimodal interactions between cerebral and intestinal functions, including the neuronal, immune signaling, and microbial-derived secretary signaling molecules, via the production of various neurotransmitters (GABA, serotonin, and dopamine) from the gut microbiota [98] (Figure 2). The gut microbiota produces neuroactive metabolites such as neurotransmitters or their precursors, which might alter the associated neurotransmitters’ concentrations [99]. This demonstrates that the neurotransmitter synthesis mechanism in the gut may impact the brain’s neuronal activity and cognitive capabilities, either directly or indirectly [100]. GBA’s vast communication system comprises a cascading program that integrates the CNS (brain and spinal cord), enteric nervous system (ENS), autonomic nervous system (ANS), and hypothalamic–pituitary–adrenal (HPA) axis [101]. The parasympathetic and sympathetic limbs of the ANS drive both afferent and efferent signals originating in the lumen and CNS, respectively [102]. The signals released from the lumen are sent to the CNS through the spinal, enteric, and vagal pathways, while signals from the CNS are directed to the intestinal wall [103].

### 5.1. Neuronal Pathway: Activation of the Vagus Nerve

The human gut has an enormous network of nerves intertwined with it, referred to as the ENS [104]. The vagus nerve, which ties the body’s visceral organs to the central nervous system, has been considered a central line of communication for the gut microbiota to monitor and control the brain and behavior. The tenth cranial nerve, the vagus nerve, conveys signals to and from the intestinal system to other organs (including the brain). It forms the most crucial part of the ANS and is the longest composite nerve that mediates communication between these two systems. The sensory neuron, which contributes 80% of the overall communication (transmits microbial-derived metabolites from gut to brain), and the motor neuron, which contributes roughly 20% of the total communication (carries the signal from the brain to the gut), are responsible for the information exchange [105]. Even though the gut microbiota can interact via endocrine and immunological routes, hijacking vagus nerve signals is undoubtedly the swiftest and most straightforward way for the microbiota to influence the brain [106]. The vagus nerve, which disseminates informative signals from the gastric mucosa to the CNS, appears to be involved in microbial communication with the brain. Indeed, no neurochemical or behavioral effects were observed in the vasectomized mice group, indicating that the vagus nerve is the pivotal modulatory fundamental communication medium between the microbiota and the brain [107]. Animal studies have established that the gut microbiota may stimulate the vagus nerve and that this activation is important in exposing effects on the brain and, as a result, behavior [108]. Early on, such information was available through the examination of pathogen-infected animals. C-fos expression has been lowered in the PVN of rats infected with *Salmonella typhimurium* after subdiaphragmatic vagotomy [109] even though *S. typhimurium* infection has been linked to intestinal inflammation; new research suggests that microbial diversity in the gastrointestinal system can effectively trigger brain circuitry regardless of whether or not there is an immune response [110]. The anxiolytic effect observed after *Bifidobacterium longum* therapy in a model of chronic colitis-related anxiety-like behavior was missing in mice vasectomized before colitis induction [111]. In studies, the gut microbiota has been found to alter these vagal-mediated effects. Certain bacterial strains have been shown to use vagus nerve signals to interact with the brain and to influence behavior [112]. For example, a subclinical dose of the diarrhea-causing bacteria *Campylobacter jejuni* enhanced anxiety-related behavior and Fos immunoreactivity in vagal afferent cell bodies and in nucleus tractus solitaries (NTS), the brain’s major projection side of the gut-related vagal afferents [113]. The vagus nerve is the chief constituent of the parasympathetic nervous system, which regulates a wide range of biological functions such as mood, immune response, metabolism, and cardiac regulation [114].

### 5.2. Microbial Signaling Molecules as a Pathway of Communication

The generation of gut microbial metabolites exhibits a vital role in the crosstalk between the gut and brain [115]. Through the immunological and endocrine pathways, gut microbiota metabolites influence the nervous system’s function [116]. The gut microbiota, for example, can emit neuro-signaling molecules such as catecholamines, GABA, melatonin, and acetylcholine (ACh) to control the CNS via the vagus nerve [117]. SCFAs (acetate, butyrate, and propionate) and other signaling molecules such as LPS, serotonin, and GABA produced by gut microbiota seem to affect the activity of the ENS that eventually modulates the afferent neural pathway, which carries the signal further to the brain. SCFAs might boost the sympathetic nervous system and mucosal serotonin secretion, modulating the brain’s memory and learning processes [118]. Apart from SFCAs, several other microbially induced compounds, such as secondary bile acids (2BAs) and tryptophan metabolites, are also involved in GBA communication [119]. These molecules predominantly transmit signals via interaction with enterochromaffin cells (ECCs), enteroendocrine cells (EECs), and the mucosal immune system. In contrast, some of them may also penetrate to enter the systemic circulation and intestinal barrier and may perhaps cross the blood–brain barrier [120].

### 5.3. Immune Signaling Pathway

The gut microbiota extensively regulates the formation, function, and maturation of the mucosal immune system, implying its possible involvement in mood and behavior control [121]. Segmented filamentous bacteria (SFB) are effective gut B and T lymphocyte stimulators in the gut [122]. The gut microbiota is also reported to use TLRs to interact with the host [123]. Interestingly, TLR-10 is expressed in various cells in the human body, such as intestinal epithelial cells, mast cells, macrophages, neutrophils, dendritic cells, lymphocytes, glial cells, and neuronal cells [124]. Microbial components can activate TLR 1–10, producing IL-1β, TNF-α, IL-6, and IL-8 [125]. Another study on TLR-4 knockout mice reported that TLR-4 facilitates inflammatory reactions and gastrointestinal problems via gut dysbiosis and leaky gut in a Gulf War disease model [126]. Additionally, the microbiota can influence hormone peptide signaling by synthesizing peptide-like antigenic proteins derived from the gut microbiota [127].

## 6. Correlation between the Gut Microbiota and Cerebral Function

The gut microbiota is known to have various health benefits in humans [128]. Moreover, recent analysis has shown that the gut microbiota can influence CNS health and diseases [129]. Thus, the exploration of the microflora residing in the brain–intestinal axis has gained popularity, garnering the concern of both gastroenterologists and neuroscientists. Recently published research articles have revealed that the gut microbiota regulates various basic neurodevelopmental and cognitive processes, such as the formation and maintenance of the integrity of the blood–brain barrier (BBB), microglia maturation, neurogenesis, and myelination, as well as the production of neurotrophins, neuromodulators, and their respective receptors [130]. These findings show that the gut microbiota plays a significant role in influencing normal human neurodevelopment [131].

The BBB could be impaired by excessive quantities of superoxide, the stimulation of MMPs, and the elevation of inflammatory mediators in the CNS [132]. Destruction of the BBB significantly increases permeability and leakage, eventually resulting in immune cell infiltration to the CNS and eventually to neuroinflammation [133]. The finding that the gastrointestinal microbiota governs neurodevelopment is absolute proof of the gut microbial population as a cerebral peacekeeper [134]. Furthermore, in rats and mice, the gut microbiota influences postnatal and adult ENS development [135]. It is worth highlighting that in mice, the gut microbiota modulates the permeability of the BBB [136]. Mono-colonization with *Bacteroides thetaiotaomicron* or *Clostridium tyrobutyricum,* as well as sodium butyrate therapy, alleviates BBB permeability in germ-free (GF) mice compared to in control animals by intensifying the biologically functional molecule of tight junction proteins [137]. Tight junction integrity is critical for sustaining BBB function [138]. These data suggest that the gut microbial community and its products may be required for appropriate BBB permeability [139].

The bacterial population residing in the ileum part of the intestine can govern both the primary immigration and the homeostatic circulation of neuroglial cells in the intestinal epithelium of mice [140]. GF mice have a much-reduced quantity and density of mucosal enteric glial cells than normal mice [141]. This discovery implies that microbiota and secretory compounds may influence gut homeostasis through enteric glial cells. Moreover, these cells in the ileum connect bacterial signals to the neurological system of the host [142].

The human body’s second genome is assumed to be gut microbiota [143]. Its makeup and variety are altered regularly depending on the circumstances. The concentration levels of numerous chemical compounds differ significantly between GF and wild-type (WT) mice in the early detection of the CNS metabolome, suggesting that the microbial community is firmly related to brain health, disease, and functions such as learning, memory, development, and behavior patterns [144]. The hippocampus serves as a learning and memory center, regulating memory encoding, spatial navigation, and memory consolidation, and is associated with mental diseases such as dementia [145]. All prior research has shown a connection between gut microbiota and hippocampal plasticity, neurochemicals, and function [146]. In a study conducted by Matsumoto and colleagues, neurotransmitters and hippocampal amino acids in GF mice at postnatal week seven were found to be significantly different from those in specific pathogen-free (SPF) mice, with decreased amounts of several amino acids, including “A, R, L, Q, I, F, V” and GABA, and increased amounts of “S”. GF mice had significantly higher levels of N-acetyl-aspartate, creatine, taurine, and lactate compared to SPF mice, while succinate levels were much lower [147]. Furthermore, the GF mice hippocampus revealed a significant increase in synaptic function via the upregulation of synapse-promoting genes, responsive microglia markers, and synaptic density, which could altogether be recovered by conventional murine microbiota or human *Bifidobacterium* species colonization, implying that *Bifidobacteria* are associated with the initiation of the functional mechanisms in the neural pathway in the hippocampus [148].

However, another study conducted by Luck’s group showed that the gut microbiota modulates brain-derived neurotrophic factor (BDNF) and cAMP response element-binding protein (CREB) in the hippocampus [149]. BDNF governs activity-mediated synaptic plasticity and psychological disorders [150], whereas CREB regulates genes involved in neural development, synaptic plasticity, acquisition, and memory [151]. Additionally, antitumor flavonoid quercetin, a phytochemical metabolite, has been demonstrated to improve cognitive performance by increasing gut microbiota and relative abundance of *Facklamia, Glutamicibacter,* and *Aerococcus,* resulting in enhanced hippocampal BDNF, thus improving learning and memory [152]. Furthermore, investigations have demonstrated that the gut microbiota modulates the shape and neurogenesis of the hippocampus. Studies on GF animals provide compelling data that the hippocampus of GF mice was noticeably larger, with pyramidal neurons that were shorter, stubby, and far less branched with mushroom spines and granule cells compared to the control mice [153].

## 7. Gut Dysbacteriosis: Consequences, Diagnostic and Therapeutic Options

### 7.1. Microbial Imbalance Leads to Several Neurological Disorders

Dysbiosis or dysbacteriosis is a disorder originating from the disruption or reorganization of the gut microbiota, resulting in a severe imbalance between beneficial and harmful microbes [154]. The following are the main factors that contribute to gut microbiota dysbiosis: (a) lack of helpful microorganisms, (b) excessive proliferation of potentially pathogenic bacteria, and (c) a decrease in the overall microbial diversity in the gut [155]. The disruption of microbial balance within the gut has been incriminated in many pathologies, including in gut-related diseases and neurodegenerative disorders (Figure 3 and Table 1) [156,157,158]. Previous studies have manifested mounting evidence that gut microbiota dysbiosis is functionally linked to brain immunological dysfunctions, contributing to poor mental health [159]. Digestion, immunopotentiation, the encouragement of microvilli development, the fermentation of dietary fibers, and pathogens required for gastrointestinal tract colonization depend on the gut microbiota [160]. According to several observational and animal studies, the gut microbiota appears to have an essential role in the neuropathogenesis of CNS disorders by GBA function alteration [161]. Dysbiosis can result in neuroinflammation by elevating bacterial metabolites and inflammatory cytokines in the gut and BBB [162]. This could play a role in the development of numerous neurodegenerative diseases, including AD, PD, MS, and amyotrophic lateral sclerosis (ALS) [163].

An SCFA-producing gut microbiota is essential to the host’s health and well-being [164]. SCFA deficiency can have various negative consequences, including inflammation, eventually contributing to neurological illnesses, namely AD, PD, and MS [165]. The comparably low abundance of these microbes might cause neurological illness etiologies or may just be an indicator of disease development [166]. These discoveries potentially pave the way for new treatment methods based on microbiota modifications. Certain compounds secreted by the gastrointestinal microbiota have subsequently been associated with cognitive and cerebrovascular illnesses [167]. New therapy techniques, including prebiotics and probiotics, may help to restore the gut microbiota, alter the gut–brain barrier, and reduce the risk of certain pathologies [168].

### 7.2. Strategies to Prevent Dysbiosis of Microbiota

The maintenance of a balanced gut microbiota is positively correlated with the host’s health status [169]. Therefore, restoring the original gut microbiota is essential to escape the cascading effect of disturbed microbial diversity, resulting in neurodegenerative disorders [170]. Several strategies have been studied and used to maintain equilibrium between beneficial and pathogenic bacteria in the gut [171]. One of the famous and effective strategies is the supplementation of probiotics and prebiotics in colonizing a healthy microbial community [172,173] (Table 2).

Probiotics are live bacteria that confer a health benefit to the host when taken in the requisite amount [184]. They can be used both to stave off the onset of dysbiosis when an individual comes across a predisposing condition (such as the inappropriate use of antibiotics, aging, chronic gut inflammation, imbalanced diet, etc.) [185] and as a curative agent to restore the healthy microorganisms to balance the ongoing condition of dysbiosis [186]. Probiotics should fall in the category of the strains resembling our normal gut microbiota, designated as GRAS (generally regarded as safe). GRAS is known as “health-friendly bacteria’’ that exhibit beneficial properties for the host’s health. These live bacteria can maintain good viability, are not toxic or pathogenic to the host, are good enough to extract nutrients from a regular diet, and do not interfere with the body’s homeostasis [187]. Some other immunological benefits of probiotics include activating the local immune cells (macrophages), helping in boosting the immune response, and modulating cytokine production [188]. Commonly used probiotics include *lactic acid bacteria, Bifidobacteria*, yeast *Saccharomyces boulardii*, *enterococci*, and Gram-negative bacteria *Escherichia coli*, [189]. *Bifidobacterium* is a well-known microorganism that produces essential vitamins, enzymes, and some acids. It also has immune activation properties resulting from lowered gut pH, thus inhibiting the growth of certain pathogens [190]. The central idea from previous studies about the benefits of probiotics leads to the conclusion that (i) probiotics can help reduce the ubiquity and severity of infectious diseases [191,192]; (ii) their unique ability to restore the gut microbiota allows them to be used as the sole treatment for many intestinal disorders and neurological disorders [193]; and (iii) if antibiotics are required, probiotics can be used in conjunction with antibiotics to decrease treatment duration and side effects [194].

Prebiotics, defined as a non-digestible fermented food ingredient, provoke the activity or growth of a number of healthy/good bacteria in the gut that play a pivotal role in human health and that naturally occur in different food products [195]. They are also widely used to rebalance the intestinal microflora. According to the International Scientific Association of Probiotics and Prebiotics (ISAPP), “dietary prebiotics are the selectively fermented ingredient that results in specific changes in the composition and/or activity of the gastrointestinal microbiota, thus conferring a benefit upon host health” [196]. By definition, they may also include a substrate consumed by the host-microbial community to confer a health benefit to the host. Prebiotic substances must have the following characteristics: they must be fermentable, capable of stimulating the activity of gut microbial diversity, and be resistant to stomach acid and mammalian enzymes [197]. Prebiotics can revitalize the gut microbiota in the intestine. Fructo-oligosaccharides (FOS) and galacto-oligosaccharide (GOS) are the most classified forms of prebiotics [198]. Some microorganisms such as *Aspergillus* sp., *Penicillium* sp., *Arthrobacter* sp., and *Aureobasidium spare* are excellent sources of a crucial key enzyme known as fructosyl-transferase (FTase) for the production of FOS [199]. Interestingly, the prebiotics fructooligosaccharides (FOS) and galactooligosaccharides (GOS) boosted *Lactobacilli* and *Bifidobacteria* development in the intestines and increased hippocampus BDNF and NR1 subunit expression in comparison to controls [200]. GOS also increases NR2A subunits in the hippocampus nucleus and NR1 expression in the frontal cortex, as well as plasma D-alanine, an NMDA receptor agonist.

Predacious employs a range of intercommunication mechanisms, one of which is predation. The term “predator” refers to bacteria that actively seek out and destroy their prey, eating their macromolecules for nourishment [201]. Predatory bacteria are important for regulating and changing bacterial populations in various settings. Even though these bacteria are nearly everywhere, only a few species have been thoroughly investigated for their potential use [202]. Because predatory bacteria are frequently smaller than their prey, they can penetrate the prey, kill it from the inside, and multiply. Interestingly, numerous predatory behaviors have emerged in the bacterial community; epibiotic predation does not require intracellular proliferation. Predatory bacteria are used to re-equilibrate a dysbiotic gut microbiota characterized by Gram-negative bacteria. In birds, the use of *B. bacteriovorus* to reestablish eubiosis in the gut microbiota has proven successful [203]. It is noteworthy that any potential therapeutic use of predatory bacteria should be based on appropriate and reliable experimental data to define the dosages to be supplied accurately. Otherwise, predatory bacteria, such as antibiotics, might become harmful in the long run if administered inappropriately.

### 7.3. Potential Biomarker for Dysbiosis and Its Implications

It is thought to be a difficult undertaking to identify “ideal biomarkers” for many diseases, including metabolic disturbances and neurological disorders [204]. Furthermore, the discovery of microbiome-based biomarkers can improve the precision of illness classification when paired with clinical data and other biomarkers [205].

One recent analysis found that urolithin detection in urine is a practical, non-invasive, and quick method that can reveal dysbiosis of the gut microbiota and intestinal inflammation in Parkinson’s disease patients [206]. *Roseburia species* is at a reduced level in Parkinson’s disease and hence can be a potential marker [207].

Several microbial metabolites may be used as predictive biomarkers to track illness conditions in association with gut microbial dysbiosis [208]. A peripheral gut microbiota-derived metabolite, indoxyl sulfate, has the potential to serve as abiomarker for metabolite profiling and diagnostic suitability for dysbiosis and neurologic signature [209]. The relative abundance of *Enterobacteriaceae*, which are expected to be used as clinical biomarkers of post-stroke cognitive impairment (PSCI), may have the ability to predict PSCI in post-stroke patients [210].

## 8. Effect of Intestinal Microbiota-Derived Metabolites on Neurological Disorders

The altered composition of the gut microbiota and its various microbial metabolites can lead to numerous brain disorders by different mechanisms. The present review article describes the effects of gut microbial-derived biomolecules on the onset and progression of most common neurological disorders, such as AD, PD, HD, and MS.

### 8.1. Aggregate-Forming Tendency of Gut Bacterial Proteins in Alzheimer’s Disease

AD is a common progressive neurodegenerative condition that causes brain shrinkage and cell death [211]. It is defined by the development of two unusual formations termed plaques and tangles, which are the main suspects in nerve cell damage [212]. Tangles are intracellular twisted fibers of a protein called tau that builds up inside cells, whereas plaques are the extracellular aggregation of a protein fragment called amyloid (A42) that builds up in the spaces between nerve cells [213].

Recent studies have suggested an association between gut microbiota-related biomolecules with the progression of Alzheimer’s. Elevated levels of numerous Gram-negative bacteria in the intestinal tract and the cerebrum of AD patients resulted in the idea of the pathogenesis and contribution of these microbial communities in respective neurodegenerative disorders [214]. LPS, a major characteristic component of Gram-negative bacteria such as *Helicobacter pylori*, *E. coli*, and *Salmonella*, was found to play a vital role in the aggregation of amyloid fibers extracellularly [215]. LPS has a potential role in the fibrillogenesis of beta-amyloid, leading to the analytical deficiency studied in in vitro conditions [216]. Meanwhile, various gut bacterial strains generate a remarkable amount of effective amyloid-like proteins. According to a discovery by Chapman’s group, extracellular fibers termed “curli”, which are formed by *E. coli* and other gut microbiota, have structural and physicochemical features with amyloids [217]. Curli was the first amyloid to be characterized in a new class of “functional” amyloids that is quickly expanding. According to scientific analysis, *Staphylococcus*, *Streptococcus*, *Mycobacteria*, *Salmonella*, *Citrobacter*, *Klebsiella*, *and Bacillus species* can construct extracellular amyloids [218].

Based on previous studies, a possible hypothesis could be drawn for the potential neurotropic role of bacterially secreted pathogenic peptides (such as cagA, vacA, and babA-*H. pylori*) and their virulent outer membrane proteins (*Hop Q*) in seeding amyloid β accumulation, which may further contribute to neuroinflammation. By employing bioinformatics tools, the aggregation tendency of these peptides could be deduced and compared with the positive control, and then inference regarding the contribution of the aggregates forming peptides could be determined [219].

### 8.2. Impact of Microbiota on the Induction of Parkinson’s Disease

PD is a neurodegenerative disorder characterized by the degeneration of dopamine-producing neurons in a specific region of the brain called the substantia nigra pars compacta and the accumulation of protein α-synuclein [220]. These pathophysiological conditions lead to movement-related disorders in Parkinson’s patients [221]. Alterations in the gut microbiota composition is strongly associated with PD. The gut microbial diversity with certain phyla, including *Firmicutes*, *Bacteroidetes*, and *Fusobacteria*, was deformed in patients with Parkinson’s symptoms compared to the control [222].

Brain behavior’s chief mediator and controller is the enrichment of specific pathogenic microbial phyla and less abundant beneficial microbial communities. SCFA, which functions as an anti-inflammatory agent and helps to prevent neuroinflammatory microenvironment creation, was found scantily in the GI tract of Parkinson’s patients due to the lower number of SCFA-producing microorganisms (*Bacteroidetes* and *Firmicutes*) [223]. A lower abundance of bacteria such as *Clostridium tyrobutyricum,* which produces high levels of butyrate, has an intended role in the progression of PD [224]. Another potential function of SCFA is maintaining the integrity of BBB and being capable of ameliorating a dysfunctional BBB in germ-free mice associated with an upregulation of tight junction protein expression [225] Additionally, in Parkinson’s patients, LPS from the innumerable gut-residing Gram-negative bacteria were found capable of crossing BBB by modulating tight junction protein expression (similar to occludin and claudin) [226]. These alterations in the expression of tight junction proteins help to accelerate the extravasation of immune cells in the brain [227], the microglial cells result in the higher production of pro-inflammatory molecules, and neuroinflammation is one of the hallmarks of initiation of neurodegenerative disorders (Figure 4) [228].

The administration of L-DOPA (levodopa), a dopamine precursor, is the most common and effective treatment to manage PD [229]. The widespread presence of *H. pylori* infection has been widely studied in this neurodegenerative disease associated with L-DOPA mal-absorption, which results in a higher motor impairment than in uninfected Parkinson’s patients [230]. Hence, eradicating *H. pylori* could lead to a remarkable refinement in motor fluctuations by improving the absorption of the Parkinson’s medication levodopa [231]. Previous studies focused on this bacterium’s ability to decrease the bioavailability of L-DOPA to the dopaminergic neuronal cells, but the mechanism is still unexplored [232].

One of the possible mechanisms could be the utilization of L-DOPA as a nutrient source by *H. pylori*, making it unavailable to be utilized by the host. Consuming L-DOPA *H. pylori* enhances its virulence, thus contributing further to aggravating this neurological disorder. L-DOPA is derived from the amino acid phenylalanine, which has a significant role in the growth and motility of bacteria [233]. As motility plays one significant role in determining a bacterium’s virulence, a hypothesis could be drawn that the consumption of L-DOPA might enhance the expression of the motility (flagellin) gene, consequently inducing motility and resulting in motility in the severity of symptoms of *H. pylori* infection in PD. Another probable mechanism is the secretion of an enzyme by the gut microbiota, which might be responsible for converting L-DOPA into an anomalous product. The biotransformation of L-DOPA into another impractical form could be one of the reasons for making it worthless for Parkinson’s therapy and reducing its availability [234].

### 8.3. Huntington’s Disease Association with Gut Dysbiosis

A trinucleotide repeat amplification in the Huntington (HTT) gene, which is distributed extensively across the brain and peripheral tissues, causes HD, a chronic neurodegenerative illness [235]. The major phyla that seem to be altered in HD patients are *Bacteroidetes* and *Actinobacteria*; meanwhile, the *Firmicutes* population was found to be diminished compared to the healthy control (HC) [236]. At the genus level, *Intestinimonas*, *Lactobacillus*, and *Bilophila* were higher in number, and comparatively, the genera *Clostridium* is less abundant than in normal individuals [237]. Huntington’s patients’ gut microbiota differed significantly from those of healthy subjects. The patients’ gut microbiota seemed to be occupied by fewer microbial species, resulting in a less diverse ecosystem [238]. A bacterial species named *Eubacterium hallii* has been linked to various clinical symptoms of Huntington’s. Symptomatic individuals with low *E. hallii* counts had more severe motor symptoms [239]. The abundance of *E. hallii* and the timing of symptom manifestation revealed a high and negative connection in pre-symptomatic patients. Intensification of the pathogenic gut microbiota (*Lactobacillus, Intestinimonas*), as well as various strains of viruses, may result in the production of harmful secretions that interfere with the production of anti-inflammatory molecules (IL-4) by specific immune cells, allowing bacteria to evade the immune response and contribute to a variety of neurological disorders [240].

A recent study on HD patients reported exclusively repressed levels of lymphokine IL-4 compared to healthy controls, whereas other cytokines remain unaltered. In the HD group, a significant correlation between systemic levels of cytokines and relative abundances of fecal microbiota has been found [241]. Interestingly, A recent study demonstrates a connection between host cytokine response in diseased and normal individuals; as a result, the level of IL-4 was found to be repressed in the HD patient compared to that of controls, while there were no significant changes in the rest of the cytokine molecules in both cases.

However, the previous study did not focus on the linkage between IL-4 secretion and the progression of HD. One of the possible reasons could be the central role of IL-4 in regulating normal brain function and a positive effect on cognitive behavior by synchronizing the activation of various cerebral growth factors such as BDNF and NGF. Simultaneously, IL-4 acts as an anti-inflammatory agent and ultimately staves off the degeneration or death of neurons [242].

### 8.4. Substantial Alteration of Human Gut Microbiota in Multiple Sclerosis

MS is a demyelinating autoimmune disease of the central nervous system (CNS) that influences the brain and spinal cord [243]. Autoimmunity against CNS antigens is hypothesized to cause immune cell infiltration, which results in CNS lesions that are diagnostic of MS [244]. Both MS and experimental autoimmune encephalomyelitis (EAE), an animal model of MS, demonstrate BBB impairment [245]. Disruption of the gut microbiota encourages permeability of the intestine, referred to as “leaky gut”, which displays one of the major contributors to the consequence of MS, an autoimmune disease [246]. The alteration of several microorganisms is associated with MS. The gut microbiota was found to be modified in different stages and types of MS, namely RRMS (Relapsing-Remitting Multiple Sclerosis), PRMS (Progressive-Relapsing Multiple Sclerosis), SPMS (Secondary Progressive Multiple Sclerosis), and PPMS (Primary Progressive Multiple Sclerosis) [247]. MS is strongly correlated with the depletion of major metabolite-producing microbes such as SCFA-producing microorganisms (Genera *Clostridium*). Butyrate- and propionate-biosynthesizing microorganisms are alleviated in RRMS compared to HC, which represents great evidence of a positive correlation between SCFA and the regression of MS [248].

Nevertheless, the mechanism of action of SCFA regarding MS plaques is underexplored. One of the possible reasons could be the neuroprotective consequence of SCFA on MS that might have a considerable role in reinstating the integrity of BBB [249]. Various SCFAs might be responsible for the higher expression level of tight junction proteins to maintain the BBB. That is how the depletion of SCFA-producing bacteria influences the cohesive property of BBB and allows the pathogenic myelin-specific T cell to enter the CNS. Additionally, SCFAs have an anti-inflammatory property that regulates the unnecessary generation of immune response by promoting Treg cell differentiation, which further reduces inflammation [250]. Another hypothesis could be drawn by the presence of detrimental bacterial and viral secretion or peptides as a result of dysbiosis, which might mimic the myelin antigen when exposed to the gut and may be responsible for the activation of myelin-specific peripheral T cells that promote their penetration to the CNS via disrupted BBB. This is how T cells could utilize one mechanism to execute the autoimmune attack on oligodendrocytes.

Rumah et al. detected *Clostridium perfringens* type B in a patient’s feces three months following the beginning of MS symptoms [251]. The epsilon toxin (ETX), released by *C. perfringens,* can pass the BBB and cause oligodendrocyte damage, suggesting a plausible mechanism for demyelination in MS. They also discovered a diminished population of *C*. *perfringens* A in the GI tract of MS patients and enhanced ETX responsiveness by ten times compared to HC. Another analysis revealed that MS patients had a considerably higher Archaeal population *(Methanobrevibacteraceae)* than the controls. *Methanobrevibacter smithii* has high immunogenicity and may cause inflammation in the patient [252]. It has also been observed that many anti-inflammatory microbes were found in decreased abundance in MS patients [253]. Significant changes in *Proteobacteria* microbiota, such as *Shigella* and *Escherichia* abundance, were also reported in pediatric MS when compared with controls.

## 9. Altered Bile Acid Profile Associates with Neurological Dysfunction

The expression of numerous bile acid receptors as well as changes in bile acid metabolism have been reported to be promising biomarkers for prognostic tools in a plethora of neurodegenerative disorders [254]. In a recent clinical study, an extensive bile acid test was carried out on the plasma of 30 healthy controls, 20 people with moderate cognitive impairment, and 30 people with clinical AD. In contrast to moderate cognitive impairment patients, AD patients had significantly higher levels of glycochenodeoxycholic acid, glycodeoxycholic acid, and glycolithocholic acid. lithocholic acid (LCA) levels were also significantly higher in AD patients than in controls. These glycine-conjugated bile acids and the presence of LCA provide valuable diagnostic biomarker features [255]. A surgical mouse prodromal PD model was used in a subsequent investigation to further explicate the existence of bile acids. Three of these acids—omega-muricholic acid, tauroursodeoxycholic acid (TUDCA), and ursodeoxycholic acid (UDCA), were found to be substantially reduced in the serum of the group treated with α-synuclein fibrils. UDCA and TUDCA, both neuroprotective secondary bile acids that can pass through the BBB, were markedly affected, with a 17- and 14-fold decrease from the control group [256].

The presence of bile acids continues to have increased relevance in maladies of the CNS beyond neurodegenerative diseases. Several bile acids have been shown to exhibit neuroprotective benefits in a variety of neurodegenerative illnesses, including AD, PD, HD, and retinal degeneration, both in cellular and animal models as well as in human clinical studies [257]. Bile acids have been studied in animal models of AD. Tauroursodeoxycholic acid (TUDCA), an endogenous bile acid, exhibits potent neuroprotective effects in a variety of disease-related experimental paradigms, including neuronal exposure to Aβ. Treatment of double transgenic mice (APP/PS1) expressing the human amyloid precursor protein with the KM670/671NL Swedish double mutation and the human presenilin 1 L166P mutation under the control of a neuron-specific promoter with 0.4% (*wt*/*wt*) of TUDCA resulted in the significantly reduced accumulation of Aβ deposits in the brain and markedly ameliorated memory deficits [258]. Additionally, the effects of bile acids on chemical and genetic models of Parkinson’s disease (PD) have been also documented. In C57BL/6 glutathione S-transferase pi (GSTP)-deficient mice, MPTP-induced degeneration of dopaminergic neurons in the nigrostriatal axis was reported to be ameliorated by TUDCA [259].

## 10. The Role of Dysbiosis in the Aging Process

A negative shift in microflora is a characteristic feature of dysbiosis. An imbalance in the body’s natural microflora can lead to physiological alterations. Geriatric alterations can also lead to changes in the macrofloral diversity and, in turn, accelerates aging-related problems [260].

Cross-sectional and longitudinal studies involving various age groups have found an alteration in gut microbial diversity with increasing age [261] (Figure 5). Researchers have found that age-related gut dysbiosis generally triggers the enhanced growth and proliferation of facultative anaerobes [262]. Meanwhile, a massive reduction in the diversity of probiotic bacteria was further followed by the altered *firmicutes*/*bacteroidetes* ratio. According to the latest analysis, the architecture and composition of the gut microbiota is a predictor of an individual’s survival. Moreover, the abundance of *Bacteroides* or a lack of originality in the microbiome is linked to shorter life expectancies and higher morbidity [263]. For instance, dysbiosis in the commensal microbial community has been shown to shorten the lifespan of Drosophila [264], but populating the gut of middle-aged African turquoise killifish with bacteria isolated from young donors led to lifetime extension and delayed behavioral deterioration [265].

One of the major risk factors contributing to the emergence of neurodegenerative disorders is aging [266]. The onset of age-related neurodegenerative diseases is facilitated by alterations in the gut microbiota [267]. An upsurge in bacterial LPS in plasma and the brain is coupled with aging [268]. The increase in LPS level is correlated with the increased expression of TLR4, myeloid differential protein-88, and the nuclear translocation of nuclear factor κB in both intestinal and brain tissues [269]. Additionally, LPS-induced ulcerative colitis mediated systemic inflammation, increasing BBB permeability, leading to inflammation in the substantia nigra region of the brain, and causing dopaminergic loss [270].

## 11. Synergetic Effect of Co-Infection and Microbial Interaction on a Neurological Disorder

The onset and progression of various infectious diseases are mainly associated with a specific pathogen, yet it is not always the case. Co-infection with two or more pathogens is now being reported to modulate the severity of neurological diseases aggressively [271]. The human host may become coinfected with multiple pathogens concurrently or may be simultaneous infected with two or more pathogens. Reactivation of latently inhabiting virulent pathogens upon infection with a new pathogen may also lead to coinfection conditions [272]. Commensal gut microbiota has been found to regulate and activate viral pathogenic genes through various mechanisms that either provoke or repress numerous diseases [273].

The gut microbiota enhances viral infection through various mechanisms, including through facilitating virus genetic recombination. Many viral genomes undergo recombination, which improves their environmental fitness and thus their infectivity [274]. After exposure to the particular commensal gut microbiota, certain RNA viruses gain an advantage in delivering their genomic content into target cells. The interaction of viruses with microbiota aids the viral entry into the host cell. It ensures genetic recombination, which further results in the generation of progeny with increased resistance to the restrictive condition. The resistive capacity of the recombinant population has a more severe effect on infectious disease outcomes. In addition, the gut microbiota also seems to enhance viral stability as it modulates viral replication [275].

Moreover, viral infection has also been associated with the disruption of the gut microbiota, resulting in colon inflammation and ultimately promoting the pathogenesis of various neurological disorders. Viral infection resulting from dysbiosis leads to the enhancement of pathogenic microbial populations such as LPS and cytokine-producing bacteria. This facilitates the aggregation of specific proteins, such as α-synuclein in the gut (ENS), that can translocate to the CNS via the vagus nerve and aggravate neurological pathologies [276]. The studies mentioned above establish that infection with a single pathogen can make the environment conducive to other pathogens’ activation. The expression of the morbific genes of co-existing pathogens may become enhanced during co-infection conditions and can be best studied in co-infection models. However, co-infection scenarios are still underexplored regarding the progression of neurodegenerative disorders. A possible hypothesis could be drawn regarding direct infection of one pathogen followed by a second infection to the neuronal cells and could be used for the study of their pathogenic gene expression, and further specific markers concerning neurological disease could be studied in the co-infected model to analyze the collaborative effect of pathogens on various neurodegenerative markers (Figure 6).

## 12. Modulation of Gut Microbiota for Neurological Disorders in the Perspective of Host-Directed Therapy: Microbiota-Targeted Technique “Fecal Microbiota Transplantation”

The proximal relationship between factors such as gut dysbacteriosis, elevated intestinal permeability, and associated neurocognitive impairment recommends that manipulating the gut microbiota may provide a promising therapeutic option in a group of neuro patients [277]. Fecal microbiota transplantation (FMT) is an approach for directly modifying the victim’s gut microbiota to normalize the proportion and to acquire therapeutic efficacy (Table 2) [278]. FMT is a biological therapy that involves the transmission of fecal matter from one fit individual to another to cure a disease [279]. It is acknowledged as the “ultimate probiotic” because it provides a significantly larger richness and variety of bacterial strains than any readily accessible probiotic. Recently, there has been a surge of interest in the potential advantages of FMT in both gastrointestinal and non-gastrointestinal diseases (Table 2) Lately, case studies of patients with MS [280], myoclonus-dystonia, autism, depression, and chronic fatigue syndrome who have been successfully treated with FMT have opened up new avenues for more promising trials in characterizing FMT as a potential therapy for such conditions. FMT has aided functional ability, favored neuronal axonal regeneration, ameliorated animal metabolic profiling and weight gain, and substantially improved gut barrier integrity and gastrointestinal motility in SCI (spinal cord injury) mice (Table 2).

A study by Yingli Jing’s group demonstrated that FMT-mediated gut microbial transformation strengthens motor, cognitive, and GI functions in SCI mice, potentially via the anti-inflammatory functions of SCFA. Elevated neurotransmitter levels of dopamine in PD mice that already had acquired FMT from healthy mice revealed the significant application of the microbiome modulation technique in various nervous system-related disorders [281].

Numerous animal studies and case reports from humans emphasize that FMT has a beneficial impact on PD, MS, AD, and stroke. The possible health benefits of FMT for patients with PD may be ameliorated by impaired α-syn accumulation in the intestinal wall and, consequently, in the brain, as a result of reduced inflammation-induced oxidative stress [282]. Greater availability and potency of levodopa following FMT with feces from donors with reduced levels of bacterial tyrosine decarboxylases in their feces could strengthen the beneficial effects on Parkinson’s symptoms [283]. A rise in Treg cell numbers after FMT in MS patients may constrict autoimmunity with demyelination and possibly the progression of the disease. Furthermore, feces from young healthy FMT donors may slow the advancement of AD by decreasing the translocation of pro-inflammatory gut microbiota from the gut to the brain and, as a result, the neuroinflammation processes facilitated by them. In conclusion, fecal microbiota transplantation can restore a drastically changed gut microbiota in neurodegenerative disorders.

## 13. Function of Modulated Microbial Communities in Healthy Aging and Rejuvenation

Age-related microbiome and gastrointestinal disorders are reversible and treatable to some extent. To change the gut microbiota and encourage healthy aging, a number of rejuvenation techniques have been used, including probiotic medication and fecal microbiota transplantation [284]. Previous research has consistently demonstrated a positive association between the abundance of beneficial microflora and centenarian longevity. For example, *Akkermansia muciniphila* stimulates the production of mucus in the gut, which is essential for maintaining intestinal integrity and other advantageous symbioses [285]. The oral treatment of *Akkermansia* also improves senescence-related phenotypes in intestinal integrity, muscle function, and immune response in aged mice and thus extends the health span, which is further supported by the understanding of the relationship between aging and the gut microbiota [286]. Additionally, the relative abundance of the butyrate producer *Oscillospira* in rejuvenated mice was dramatically raised by all rejuvenation techniques [287].

Besides probiotics, the consumption of some of the chemical compounds has been proven to be helpful for the maintenance of gut microbial health [288]. *N*-Acetylcysteine (NAC) is an FDA-approved drug primarily associated with its anti-inflammatory and antioxidant activity, which supports the maintenance of a cellular redox imbalance [289]. It is a potential drug to prevent glucose metabolic disturbances by reshaping the structure of the gut microbiota.

In HFD-fed mice, NAC significantly ameliorates gut barrier disruption, glucose intolerance, and inflammatory responses [290]. The potential regulatory mechanism may help to reshape the altered gut microbial structure by intensifying the growth of beneficial bacteria such as *Akkermansia*, *Bifidobacterium*, *Lactobacillus*, and *Allobaculum* and by reducing the abundances of harmful bacteria such as *Desulfovibrio* and *Blautia*, which are concurrent with the repair of disrupted metabolic pathways. These investigations reveal the prominent role of NAC in alleviating gut dysbiosis and associated inflammation [291].

## 14. Conclusions

The architecture and composition of the gut microbiota undergo significant changes throughout time, and these alterations are typically associated with or followed by adverse health consequences. Numerous factors, such as the way of life, stress, nutritional challenges, antibiotics, and the aging process may alter the gut microbiota. It has been established that strategies to counteract these detrimental fluctuations contribute to alleviating symptoms and recovery from certain conditions. The gut–brain axis is an important field of study, with several studies associating changes in gut microbiota composition with a range of clinical disorders. The discovery of a biological connection between the microbiota, immune signaling, and the CNS shows that the microbiota’s microbial metabolites or systemic signals may alter neurological and immune function in the brain. Cancer, autoimmune illnesses, and neurological conditions such as MS, PD, and AD have all been linked to dysbiosis in the gut microbiota. Bacteria impact the immunological responses of the host in part by producing metabolites. By regulating and reactivating each other’s pathogenic genes, synergistic interactions between many pathogens may play a role in the onset and progression of various neurodegenerative disorders. The dysbiosis of the gut microbiota is caused by a rise in pathogenic microorganisms, which may be restored by probiotics, prebiotics, predatory bacteria, and fecal microbial transplantation from healthy individuals. These methods may assist in the battle against neurological symptoms resulting from microbial dysbiosis. Furthermore, the discovery of microbiome-based biomarkers can improve the precision of illness, including metabolic disturbances and neurological disorders. In conclusion, the microbial population inhabiting and dwelling on the human body serves a significant function in preserving the human host’s health. Any imbalance between “good” and “bad” microflora may cause illness and reduce the quality of life of a person. Therefore, it is essential that the function of microbiota in many diseases, particularly neurological disorders, be highlighted and thoroughly researched.

## Figures and Tables

**Figure 1 metabolites-12-01064-f001:**
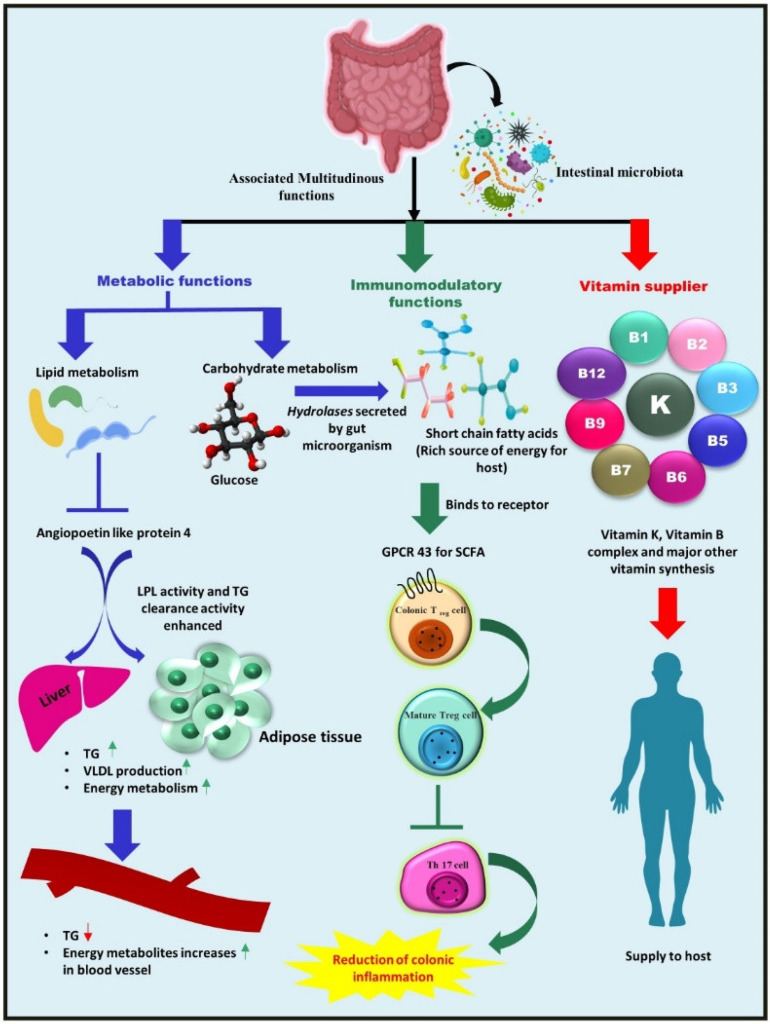
Summary of the gut microbiota-associated functioning in the human body. The gut microbiota leads to enhanced lipid clearance by repressing angiopoietin-like protein 4 (Angptl4), an inhibitor of LPL, due to which energy metabolites are elevated in the serum, which is demonstrated by an increase in associated genes in the liver transcriptome. Triglyceride levels in the serum are reduced while they rise in adipose tissue and in the liver. Dietary carbohydrates are metabolized into SCFAs, a rich energy source for the host by colonic bacteria by employing a special class of enzyme, hydrolases. Furthermore, these SCFAs, especially butyrate, can provoke the GPCR 43 expressed by intestinal epithelial cells and regulate the development, differentiation, and maturation of Treg cells via epigenetic regulation, resulting in the inhibition of Th17 cell development and the reduction of colonic inflammation. Selected gut microbiota can also act as vitamin suppliers to the host, as they synthesize vitamin B complexes and vitamin K and supplies them to the host.

**Figure 2 metabolites-12-01064-f002:**
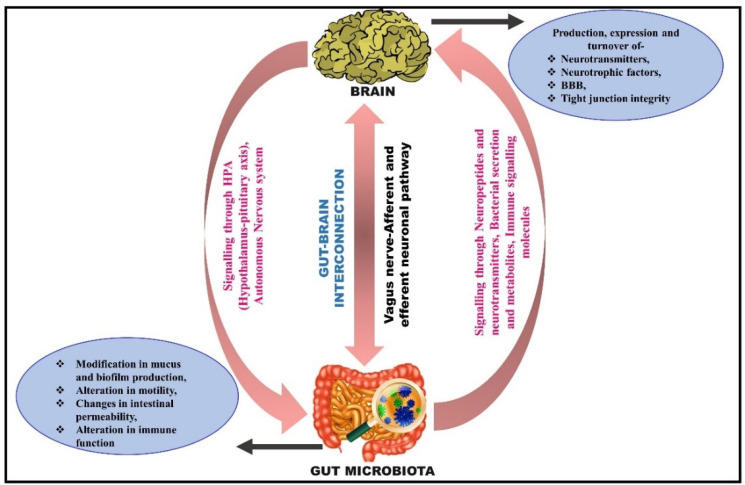
Illustrative diagram depicting the interconnection between gut and brain. The proposed bidirectional communication is firmly affected by various pathways, including the autonomic nervous system (ANS), hypothalamic–pituitary–adrenal (HPA), immune pathways, enteric nervous system (ENS), endocrine pathways, and neural pathways. Gut microbiota can produce microbial metabolites that activate the neuroenteric plexus, stimulate neuropeptide production in the brain, and increase gut–blood barrier and blood–brain barrier (BBB) permeability. The brain releases molecules that stimulate the function of the gut and neuroendocrine plexus.

**Figure 3 metabolites-12-01064-f003:**
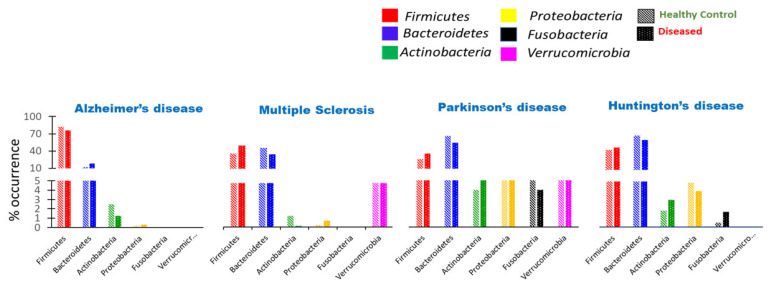
Variation in the composition of microbial diversity at the phylum level in numerous neurological disorders (AD, MS, PD, and HD). Changes in the occurrence of gut microbial population, dominant gut phyla, including *Actinobacteria*, *Bacteroidetes*, *Firmicutes*, *Proteobacteria*, *Verrucomicrobia*, and *Fusobacteria,* in neurological illnesses compared to in healthy controls.

**Figure 4 metabolites-12-01064-f004:**
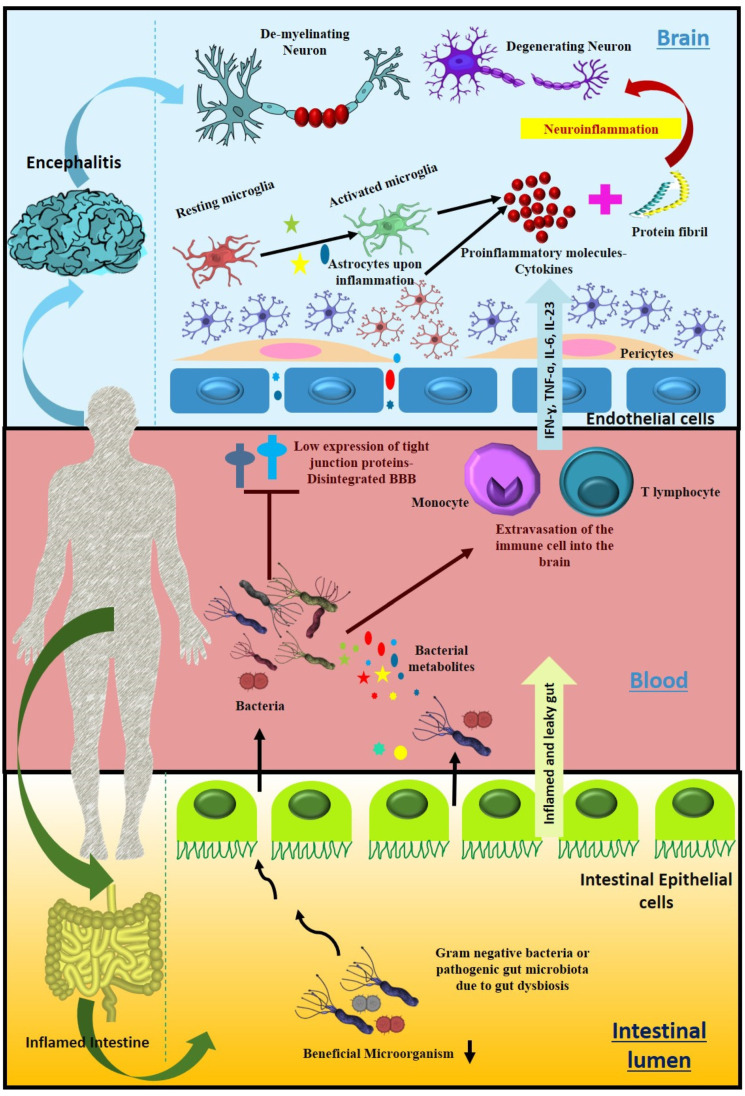
Probable mechanism of gut microbial population and their metabolites to cause neuroinflammation. Intensification and accumulation of pathogenic gut microbiota due to dysbacteriosis lead to the formation of an inflamed and leaky gut. Microbes and their associated metabolites can enter the peripheral circulatory system, where they further lower the expression of tight junction proteins in brain endothelial cells and lead to the disintegration of the blood–brain barrier that promotes the entrance of pathogenic microbes, some of them carrying prions into the brain. Meanwhile, microbial metabolites stimulate the extravasation of immune cells into the brain, all of which together trigger cytokine storm, the cause of neuroinflammation, one of the important hallmarks of neurological disorders characterized by degenerated and demyelinated neurons.

**Figure 5 metabolites-12-01064-f005:**
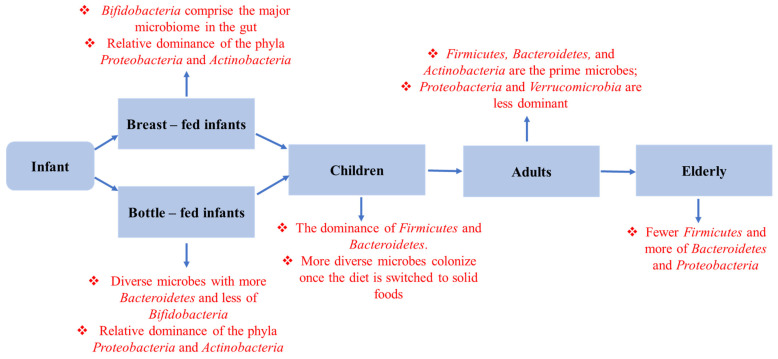
Composition of gut microbiota in the healthy aging process. The age-related sequential change in gut microbiota composition and metabolic function from infant to centenarian.

**Figure 6 metabolites-12-01064-f006:**
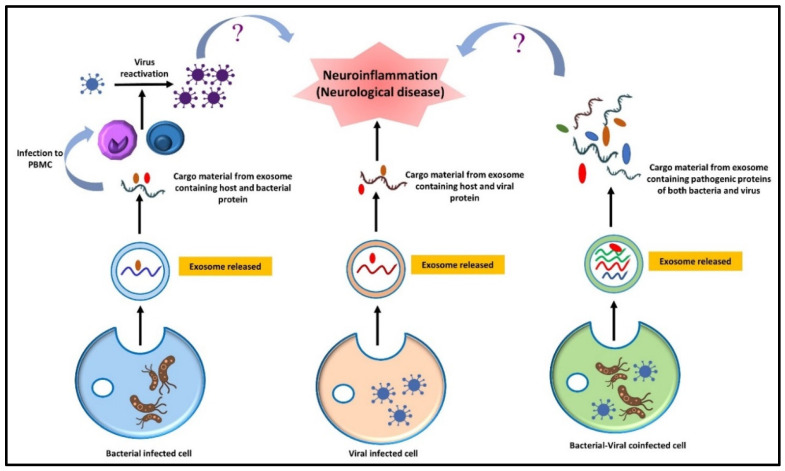
Schematic diagram depicting the effect of a lone bacterial or viral infection and pathogenic coinfection in engendering neuroinflammation. The role of extracellular vesicles from bacteria-infected cells in virus reactivation via PBMC in contributing to neuroinflammation has to be explored. Meanwhile, the analysis of the combined effect of pathogens on various neurodegenerative markers and inflammation could be contemplated in the co-infection models. NB:? signifies the nonavailability of direct evidence and study needs to be done in this aspect.

**Table 1 metabolites-12-01064-t001:** Variation in the composition of microbial diversity at the genus level in numerous neurological disorders. Abundance or scarceness of specific genera in a particular neurological disease compared to in healthy controls, where ↑ represents higher in number compared to healthy ones, ↓ represents lower in number compared to healthy ones, and-represents not reported yet.

GENUS	Alzheimer’s Disease	Parkinson’s Disease	Huntington’s Disease	Multiple Sclerosis
*Bifidobacterium*	↓	↑	↑	-
*Clostridium*	↓	↓	↓	↑
*Dialister*	↓	-	-	-
*Turicibacter*	↓	-	-	-
*Bacteroides*	↑	↓	↓	-
*Blautia*	↑	↓	-	↑
*Bilophila*	↑	-	-	-
*Lactobacillus*	-	↓	-	↓
*Faecalibacterium*	-	↓	↑	-
*Coprococcus*	-	↓	-	-
*Prevotella*	-	↓	↑	↓
*Akkermansia*	-	↑	-	↑
*Methanobrevibacter*	-	-	-	↑
*Butyricimonas*	-	-	-	↓
*Collinsella*	-	-	-	↓
*Slackia*	-	-	-	↓
*Megamonas*	-	-	↓	-
*Gemmiger*	-	-	↑	-
*Allistipes*	↑	-	↓	-

**Table 2 metabolites-12-01064-t002:** **Modulation of the gut microbiota as a therapeutic approach for several neurological deformities.** Direct modulation of the gut microbiota can be a potential therapeutic target for treating neurological disorders. The table summarizes recent research findings on the reformation of the gut microbiota by probiotic therapies and fecal microbiota transplantation to overcome the neuropathologic condition.

S. No.	Therapeutic Approach	Phylum/Genus or the Name of Particular Bacteria	Neurological Disorder	References
**1.**	Recolonization of beneficial bacteria	Oral administration of *Bacteroides fragilis*	Experimental autoimmune encephalomyelitis	[174]
**2.**	Fecal microbiota transplantation	Fecal microbiota suspension was injected through a TET tube.	Parkinson’s disease	[175]
**3.**	Bacteriotherapy	Transcolonoscopic infusion of 13 non-pathogenic enteric bacteria	Chronic Fatigue Syndrome (CFS)	[176]
**4.**	Administration of human commensal bacteria	*Bacteroides fragilis*	MIA mouse model of ASD (autism spectrum disorder)	[177]
**5.**	Probiotic supplementation	*Lactobacillus acidophilus*, L. *fermentum*, *Bifidobacterium lactis*	Alzheimer’s disease	[178]
**6.**	Fecal microbiota transplantation	FMT from healthy uninjured mice	Spinal cord injury (SCI) mice model	[179]
**7.**	Probiotic supplementation	*Bifidobacterium bifidum BGN4 and *Bifidobacterium longum* BORI*	Alzheimer’s disease	[180]
**8.**	Probiotic supplementation	*Bifidobacterium infantis*	Rat maternal separation (MS) model of depression	[181]
**9.**	Fecal microbiota transplantation	Fecal suspension injected into the colon through the catheters	Traumatic brain injury (TBI) in male Sprague Dawley rats	[182]
**10.**	Probiotic supplementation	*Streptococcus thermophilus*, *Bifidobacterium lactis*, *Lactobacillus acidophilus*, *Lactobacillus helveticus*,	Alzheimer’s disease triple-transgenic mice	[183]

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
