# Peer review of "Dysbiosis of Gut Microbiota from the Perspective of the Gut–Brain Axis: Role in the Provocation of Neurological Disorders"

_metabolites, 2022, doi:10.3390/metabo12111064_

Round 1

Reviewer 1 Report

Kandpal et al. have presented a paper on the role of gut dysbiosis in the onset of neurological disorders. The paper is well-structured, has a decent size according to the subject, and touches on the most important aspects of the topic. There are, however, some required improvements in order for the manuscript to be considered for publication:

- the main issue is that the paper is heavily under-referenced and multiple paragraphs and phrases are missing citations; this is a significant issue that should be corrected throughout the paper; statements such as "a study published by..." end without citing the paper which is unacceptable; some of the instances: lines 57-59, 114-116, 116-119, 131-135, 138-142, 176-179, and so on...; please address this issue and make sure that no statements are made without being backed up by the adequate, relevant citations; for a review of this size, one would expect around 300 total references

- the introduction should also present the speculated connection between the gut microbiota and neurological disease that is presented in the abstract; on the other hand, features such as enumerating the segments of the GI tract are unnecessary

- perhaps the introduction could also touch on the gut-lung-skin axis or other interferences and/or connections between these epithelial structures

- the authors describe the "robust" microbiota in the heading of the second chapter - what does that mean? the term is not used in any other part of the paper

- the title of Figure 1 should clearly refer to "gut" or "intestinal" microbiota

- Figure 3 reproduces data of another article(s)? is the figure reproduced with copyright? is it necessary to this manuscript?

- how is chapter 7 different from chapter 8? chapter 7 presents the relationship between dysbiosis and neurological ilness while chapter 8 presents the effects of altered composition of gut microbiota on brain disorders, therefore essentially the same; please adapt the titles and/or the content for better readability and clarity for the readers

- what "manipulation" is presented in chapter 8.4? it is unclear from the paragraphs of this section that any interventional studies were performed to demonstrate consequences on MS of shifting the microbiota; please clarify

- for chapters 10 and 11 it would be beneficial to add a table listing the  (relevant) studies on the modulation of microbiota and the impact on the neurological disorders

- author contributions should be rewritten in accordance with the instructions for authors

- the abbreviations in the abstract are unnecessary as the terms appear only once

Respectfully submitted,

Author Response

Reviewer 1 comment 

References

  • The main issue is that the paper is heavily under-referenced and multiple paragraphs and phrases are missing citations; this is a significant issue that should be corrected throughout the paper; statements such as "a study published by..." end without citing the paper which is unacceptable; some of the instances: lines 57-59, 114-116, 116-119, 131-135, 138-142, 176-179, and so on...; please address this issue and make sure that no statements are made without being backed up by the adequate, relevant citations; for a review of this size, one would expect around 300 total references.

Ans- Thank you for your valuable suggestion. New relevant references have been added to the current form of the manuscript. The current form of the manuscript contains 271 references in total.

Introduction

  • The introduction should also present the speculated connection between the gut microbiota and neurological disease that is presented in the abstract; on the other hand, features such as enumerating the segments of the GI tract are unnecessary

Ans- Thank you for your insightful suggestion, in the current form of manuscript we have explained the connection between gut microbiota and neurological disease in the Introduction section with an appropriate citation which can be found in lines 103-108. Also, the segments of the GI tract have been removed from line 99 in the current form of the manuscript as suggested by the respected Reviewer.

Introduction

  • Perhaps the introduction could also touch on the gut-lung-skin axis or other interferences and/or connections between these epithelial structures

Ans- Thank you for your valuable suggestion. Explaining the interactions between different epithelial structures with gut dysbiosis is an interesting aspect. However, the current review is primarily focused on gut dysbiosis and its association with neurological modalities. Hence, we have discussed the Gut-brain axis in only.

The authors describe the "robust" microbiota in the heading of the second chapter - what does that mean? the term is not used in any other part of the paper

Ans- Thank you for your comment. The word "robust" has been removed from the title of the second chapter in the current form of manuscript.

  • The title of Figure 1 should clearly refer to "gut" or "intestinal" microbiota

Ans- Thank you for your suggestion. We have corrected the figure legend of figure 1 by adding the “ gut microbiota”.

  • Figure 3 reproduces data of another article(s)? is the figure reproduced with copyright? is it necessary to this manuscript?

Ans- Thank you so much for your comment. Yes, figure 3 has been reproduced by collecting information from different previously published manuscripts, where data was present in a different form. Here we have compiled this data and presented it in a figure form. Hence there is no copyright violation as we are not recreating the image. Appropriate citations has been also done for the collected data.

  • How is chapter 7 different from chapter 8? chapter 7 presents the relationship between dysbiosis and neurological ilness while chapter 8 presents the effects of altered composition of gut microbiota on brain disorders, therefore essentially the same; please adapt the titles and/or the content for better readability and clarity for the readers

Ans- Thank you for your valuable suggestion. In our manuscript chapter 7 deals with the changes in the microbial composition of the gut and its association with neurological disorders. Moreover, in chapter 8 we discussed specific pathogenic factors and their possible mechanism in the development of specific neurological disorders like AD, PD, HD and MS.

  • What "manipulation" is presented in chapter 8.4? it is unclear from the paragraphs of this section that any interventional studies were performed to demonstrate consequences on MS of shifting the microbiota; please clarify

Ans- Thank you for your comment. Word "manipulation" has been replaced by “Substantial alteration”. There is limited evidence related to the effect of gut dysbiosis in the onset of Multiple Sclerosis. However, there are evidences regarding the altered function of gut microbiota during the progression of Multiple Sclerosis. So in our manuscript, we have compiled all the possible information related to MS and its association with gut dysbiosis.

  • For chapters 10 and 11 it would be beneficial to add a table listing the  (relevant) studies on the modulation of microbiota and the impact on the neurological disorders

Ans. Thank you for your valuable suggestion. We have added a table entitled “Modulation of the Gut Microbiota as a Therapeutic Approach for several neurological deformities” in chapter 11 and Page no. 27.

  • Author contributions should be rewritten in accordance with the instructions for authors

Ans. Thank you for your correction. The author’s contribution has been rewritten in accordance with the instructions for authors

  • The abbreviations in the abstract are unnecessary as the terms appear only once

Ans. Thank you for your insightful suggestion. The abbreviations have been removed from the abstract section.

Reviewer 2 Report

Journal: Metabolites-1971647

Type of manuscript: Review

Title: Gut dysbiosis from the perspective of gut-brain axis: Role in the provocation of neurological disorders

The authors reviewed the gut-brain axis showing the bidirectional communication network connecting the gastrointestinal tract and CNS. Specifically, they examined the increasing literature about the relationship between intestinal dysbiosis and the emergence of neurological diseases, such as Alzheimer's disease (AD), Parkinson's disease (PD), Huntington's disease (HD), and multiple sclerosis (MS).

The manuscript is well presented and discussed, suggesting the beneficial effects of restoring the human gut flora through probiotics, prebiotics, predatory bacteria, and even fecal microbial transplantation from healthy individuals. However, there are some considerations the authors have to address.

1.- I would appreciate the inclusion of a brief subheading about the role of dysbiosis in the aging process. I think the inclusion would improve the present excellent review.

2.- It is well known that aging is the principal risk factor for neurodegenerative diseases. Therefore I have some questions about the role of aging-associated changes in the gut microbiota and its role in neurodegeneration. What is the function of modifying microbial communities in healthy aging and rejuvenation? Are there some models? Please see some essential references: Shin et al., Microbiome (2021) 9, 240. Shukla et al. iScience (2021) 24,7 102703.

3.- An exciting issue in intestinal homeostasis is the implication of reactive oxygen species (ROS) generated in the colonic mucosa resulting from gut dysbiosis. ROS affects signal transduction pathways by the reversible oxidation of sensitive-cysteine residues in specific cysteine-containing proteins, usually regulatory enzymes. The current paradigm holds that redox signaling occurs via cysteine redox proteome. What is the role, if any, of microbial-induced redox-dependent intestinal signaling in gut dysbiosis? Could this redox dysregulation alter the gut-brain axis? What could be the potential role of modifying the gut cysteine redox proteome in aging and neurodegenerative diseases?

4.-  N-Acetylcysteine (NAC) is a safe drug that has been implicated in the prevention and treatment of metabolic disturbances. Do the authors know any beneficial role of NAC in regulating gut microbiota?

5.- Is there any potential biomarker of gut dysbiosis in neurodegenerative diseases? Can some of these biomarkers help to improve the diagnosis, prevention, and treating disease progression in microbiota-based interventions?

Author Response

  • I would appreciate the inclusion of a brief subheading about the role of dysbiosis in the aging process. I think the inclusion would improve the present excellent review.

Ans. We are grateful for your valuable suggestion to improve the current manuscript immensely. As suggested, we have included a new section entitled as “The role of dysbiosis in the ageing process” in section 9 and page 23 in which we have included the importance of gut microbial diversity in gerontology and its implications in the associated neurological complications.

  • It is well known that aging is the principal risk factor for neurodegenerative diseases. Therefore I have some questions about the role of aging-associated changes in the gut microbiota and its role in neurodegeneration. What is the function of modifying microbial communities in healthy aging and rejuvenation? Are there some models? Please see some essential references: Shin et al., Microbiome (2021) 9, 240. Shukla et al. iScience (2021) 24,7 102703.

Ans. Thank you for your insightful suggestion. As per the suggestion, we have included a new section “The role of dysbiosis in the ageing process” in section 9 and pages (23 and 24) in which we have also discussed the role of aging-associated changes in the gut microbiota and its role in neurodegeneration as per the current available literature. Maintenance of good microbial diversity is associated with healthy ageing. Further techniques like FMT (Faecal microbiota transplantation) and probiotic medication has been used for rejuvenation. We have included a sub-section under the heading dedicated “Function of modulated microbial communities in healthy ageing and rejuvenation” in section 13 and page 30 in which we have discussed the role of modulated microbiota to encourage healthy ageing.

  • An exciting issue in intestinal homeostasis is the implication of reactive oxygen species (ROS) generated in the colonic mucosa resulting from gut dysbiosis. ROS affects signal transduction pathways by the reversible oxidation of sensitive-cysteine residues in specific cysteine-containing proteins, usually regulatory enzymes. The current paradigm holds that redox signaling occurs via cysteine redox proteome. What is the role, if any, of microbial-induced redox-dependent intestinal signaling in gut dysbiosis? Could this redox dysregulation alter the gut-brain axis? What could be the potential role of modifying the gut cysteine redox proteome in aging and neurodegenerative diseases?

Ans. Thank you for your valuable comment. As per our understanding, there are limited pieces of evidence which suggest the direct relation of cysteine redox proteome with gut dysbiosis. Further, the role of cysteine redox proteome in ageing and neurological disorders still warranted and is currently out of scope for the current form of manuscript.

  • N-Acetylcysteine (NAC) is a safe drug that has been implicated in the prevention and treatment of metabolic disturbances. Do the authors know any beneficial role of NAC in regulating gut microbiota?

Ans. Thank you for your insightful concern. As per the suggestion, we have included the section entitled “Function of modulated microbial communities in healthy ageing and rejuvenation” in section 13 and page 30 in which we have discussed the beneficial role of NAC in regulating gut microbiota. It has been found to be a potential drug to prevent glucose metabolic disturbances by reshaping the structure of the gut microbiota.

  • Is there any potential biomarker of gut dysbiosis in neurodegenerative diseases? Can some of these biomarkers help to improve the diagnosis, prevention, and treating disease progression in microbiota-based interventions?

Ans. Thank you for your precious suggestion. There are some potential biomarkers of gut dysbiosis in neurological disorders and this field is in the process of exploration. Still, we have included the available information in a sub-section under the heading dedicated “Potential biomarker for dysbiosis and its implications” in section 14 and page 30.

Note: As per the suggestions by respected reviewers, the revisions have been made and highlighted in the current form of the manuscript with yellow colour in text, table, and figure.

Reviewer 3 Report

Reviewer comments for metabolites-1971647

This manuscript summarized the relationship between gut dysbiosis and neurological disorders via the gut-brain axis. This review is informative and well-written. However, several minor issues exist should be revised, details as follows:

1.      Even though acronyms were mentioned in the abstract, the full name of the term and acronyms should be stated again when they first appeared in the manuscript, such as AD, PD…

2.      The present study mainly focused on gut microbiota and neurological disorders, but the authors did not mention gut microbiota in the title, only using gut dysbiosis. We recommend revising the title, e.g. “dysbiosis of gut microbiome” or “imbalance of gut microbiota” etc.

3.      In the whole manuscript, several kinds of expression for microorganisms in the gut were used, like “gut microbiota”, “gut microbiome”, “gut flora”, “gut bacteria”, “intestinal flora”, in and so on. If possible, unified or reduced expressions would be recommended.

4.      If the contents regarding fungi, mycobiota and virome is few, we suggest removing line 97-102. 

Author Response

Reviewer 3 comment 

  • Even though acronyms were mentioned in the abstract, the full name of the term and acronyms should be stated again when they first appeared in the manuscript, such as AD, PD…

Ans. Thank you for your useful suggestion. Acronyms and their full term have been stated again where they appeared in the current form of the manuscript.

  • The present study mainly focused on gut microbiota and neurological disorders, but the authors did not mention gut microbiota in the title, only using gut dysbiosis. We recommend revising the title, e.g. “dysbiosis of gut microbiome” or “imbalance of gut microbiota” etc.

Ans. Thank you for your insightful suggestion. The title of the current manuscript has been changed as suggested by the respected reviewer.

  • In the whole manuscript, several kinds of expression for microorganisms in the gut were used, like “gut microbiota”, “gut microbiome”, “gut flora”, “gut bacteria”, “intestinal flora”, in and so on. If possible, unified or reduced expressions would be recommended.

Ans. Thank you for your valuable suggestion. We have unified the term throughout the current form of the manuscript.

  • If the contents regarding fungi, mycobiota and virome is few, we suggest removing line 97-102. 

Ans. Thank you for your comment. As per the suggestion, we have removed the content regarding fungi, mycobiota and viromes in modulating intestinal physiology, regulating and maintaining intestinal homeostasis, and influencing systemic immunity in the current form of manuscript.

Round 2

Reviewer 1 Report

The authors have significantly improved on the paper and have addressed all previously mentioned concerns.

Author Response

                                                           Pointwise Response

Editor 1  comment 

  1. Although the authors have answered most questions, I found that they have a few sections that are not relevant to gut dysbiosis, so I have highlighted those. You will also find a few comments throughout the text (attached pdf). An important topic that the authors did not even describe is “bile acids and neurodegeneration”. Bile acids have been studied with respect to AD and PD and are key components of the gut microbiome.

Ans 1- Thank you for your valuable suggestion. New section regarding Role of gut microbiota in bile acid metabolism has been incorporated in subsection 2.1. of chapter 2 in line number 129-141 and highlighted with blue colour in the current form of manuscript. Additionally, a new section entitled “Altered bile acid profile associates with Neurological dysfunction” has been also added as per the suggestion in chapter 9, Page no. 19 and highlighted with blue colour in the current form of manuscript.

  1. The entire sentence copied from the description on this page: https://www.mdpi.com/journal/ijms/special_issues/mechanisms_neurodegeneration

Ans 2- Thank you for your valuable comment. As per the suggestion the sentence has been reframed, can be found in line 22 and 23 and highlighted with blue colour.

  1. This section mentions host factors controlling gut microbiota, but it's not linking to any neurological conditions.

Ans 3- Thank you for your precious suggestion. In this section, we have tried to incorporate the host factors influencing composition of gut microbiota and the consequences of modified gut microbiota has been already described in the chapter 7 and chapter 8 of manuscript with regards to neurological disorder. Additionally, as per the suggestion from editior 2, this section has been shifted after chapter 2 and it is now chapter 3 and its subheading into subsection “3.1, 3.2, 3.3, 3.4. Revised section has been highlighted with blue colour.

  1. Need to cite the reference from where the data was obtained for this plot. I checked reference 148, and it has data for AD and controls only. I am curious to know the from where the data for MS, PD and HD came.

Ans 4- Thank you for your insightful suggestion. The newer references related to the data of MS, PD and HD has been added. Refernce number includes- 156,157,158 and 26. and has been highlighted with blue colour.

Editor 2  comment 

  1. Species names are irregularly italicized in the manuscript. To name a few, in lines 135, and 136 there are names that are not italicized, whereas lines 93, and 95 are italicized.

Ans 1. Thank you for your valuable suggestion. Species name has been italicized throughout the current form of manuscript.

  1. Please simplify the name of section 2– need to read the subheadings to understand what to expect in the content. “Composition of gut microbiota-associated multitudinous function in a healthy and diseased state”

Ans 2. Thank you for your insightful suggestion. As per the suggestion we have reframed the title of chapter 2 and named as “Composition of gut microbiota and its associated multifarious function to the host heath” and has been highlighted with blue colour.

  1. Section 6 “Host factors controlling gut microbiota“ seems disconnected to the previous and next sections. I would recommend moving it before any brain-gut connection is mentioned, perhaps after section 2.

Ans 3. Thank you for your valuable suggestion. We have shifted the section “Host factors controlling gut microbiota” after chapter 2 and it is now chapter 3 and its subheading into subsection “3.1, 3.2, 3.3, 3.4. Revised section has been highlighted with blue colour.

  1. Section 7, 11, and 14 belong together in my opinion

Ans 4. Thank you for your insightful suggestion. We have merge these 3 sections into 1 heading and its respective sub-section and highlighted with blue colour.

  1. Gut dysbacteriosis: Consequences, Diagnostic and therapeutic options

7.1. Microbial imbalance leads to several neurological disorders

7.2. Strategies to prevent dysbiosis of microbiota

7.3. Potential biomarker for dysbiosis and its implications
